# More than just visits: Timing, frequency, and determinants of effective antenatal care in Bangladesh - BDHS 2007 to 2017-18

Md. Hasibul Islam Jitu[1]*, Awan Afiaz[2], Raaj Kishore Biswas[3]

1 Institute of Statistical Research and Training, University of Dhaka, Dhaka, Bangladesh, 2 Department of Biostatistics, University of Washington, Seattle, Washington, United States of America, 3 School of Health Sciences, Faculty of Medicine and Health, The University of Sydney, New South Wales, Australia

* mhjitu@isrt.ac.bd

## Abstract

### Background

Timely initiation and adequate number of antenatal care (ANC) visits are crucial for ensuring the health and well-being of both pregnant women and their unborn children. Despite recent progress, Bangladesh continues to face challenges in achieving sustainable development goal (SDG-3) related to maternal and neonatal health. This study examines the factors contributing to delayed initiation and a low number of ANC visits, while also evaluating the association between the timing and overall number of ANC visits.

### Data

Nationally representative data from the Bangladesh Demographic and Health Surveys (BDHS) conducted in 2007 (n = 3050) and 2017–18 (n = 4544) on women aged 15–49 years.

### Methods

We investigated two binary outcome variables: late ANC, defined as the initiation of ANC visits after 12 weeks of gestation, and low ANC, defined as having less than four ANC visits. Geospatial mapping was employed to visualize spatial patterns, followed by survey-weighted logistic regression to identify risk factors associated with late initiation of ANC and low ANC visit frequency. Additionally, classification tree analysis was utilized to explore interactions between predictors and outcomes.

### Results

Logistic regression modeling revealed that late ANC was associated with a more than fourfold increase in the odds of having fewer than four ANC visits (AOR: 4.60 [95% CI: 3.69–5.73] in 2007 and AOR: 4.68 [95% CI: 4.00–5.48] in 2017–18). Classification

**Data availability statement:** The secondary datasets analyzed in this study are available upon request from the Demographic and Health Surveys (DHS) Program. Access to the data requires registration and approval through the DHS website: https://dhsprogram.com/data/available-datasets. The authors do not have the authority to share the data directly.

**Funding:** The author(s) received no specific funding for this work.

**Competing interests:** The authors have declared that no competing interests exist.

tree analysis further confirmed that late ANC initiation was the most critical predictor of total number of ANC attendance, underscoring the necessity of early ANC initiation to ensure sufficient coverage.

## Conclusion

Early initiation of ANC is essential for achieving an adequate number of ANC visits. Notably, the same set of sociodemographic factors remained statistically significant predictors in both 2007 and 2017, highlighting the persistent nature of these disparities and underscoring the urgent need for targeted policies and health interventions.

## Introduction

Timely and adequate access to antenatal care (ANC) is essential for reducing the risks associated with pregnancy and childbirth, thereby improving maternal and neonatal health outcomes [1]. When provided by medical professionals throughout pregnancy, from conception to labor, ANC encompasses comprehensive services, including risk assessment, disease prevention, treatment, and health education [2,3]. In Bangladesh, however, a significant proportion of women delay the initiation of ANC until later stages of pregnancy and fail to attend the recommended number of ANC visits [4]. Therefore, improving the timely initiation and adequacy of ANC is imperative for advancing maternal healthcare in Bangladesh, as it plays a pivotal role in reducing maternal and neonatal risks and improving overall health outcomes.

The timely initiation of ANC in the first trimester is critical for optimizing maternal and fetal health outcomes, as it provides an opportunity for early intervention, risk assessment, and the implementation of preventive measures that support a healthier pregnancy and childbirth [2]. Achieving Sustainable Development Goal 3 requires prioritizing timely and adequate ANC visits, as they are fundamental to reducing maternal mortality (target 3.1) and preventing neonatal and under-5 mortality (target 3.2) [5]. Furthermore, the Global Strategy for Women's, Children's, and Adolescents' Health (2016–2030) seeks to eliminate stillbirths and other preventable deaths among women, children, and adolescents by 2030, emphasizing the critical role of comprehensive and timely healthcare interventions [6]. Timely and sufficient ANC visits can play an important role in achieving this goal.

Prior to 2016, the WHO recommended at least four ANC visits during pregnancy (the FANC model) [7]. The revised 2016 WHO guidelines increased the requirement to at least eight visits [3]. However, Bangladesh's current national guideline still recommends four visits [4]. The first ANC visit should occur by 12 weeks (WHO 2016 guideline), 8–12 weeks (FANC model), and by 16 weeks (Bangladesh national guideline) [3,4,7].

Previous studies have demonstrated that timely and adequate ANC visits are strongly associated with reduced risks of low birthweight, stillbirth, anemia, perinatal and neonatal mortality, as well as maternal mortality [8–10]. Moreover, routine ANC visits offer valuable opportunities for screening non-communicable diseases, such as

diabetes, and provide essential guidance on lifestyle modifications to mitigate risks associated with drug and alcohol misuse, smoking, obesity, and malnutrition [2]. However, home births, often assisted by traditional birth attendants (TBAs), continue to be prevalent in Bangladesh due to economic constraints, traditional beliefs, religious factors, limited access to healthcare facilities, and a lack of awareness about alternative options [11]. These factors may contribute to reduced access to essential ANC visits. Additionally owning mobile phones also significantly plays a critical role in improving maternal health outcomes in low- and middle-income countries (LMICs) like Bangladesh. Studies have found that mobile phones help better access to health information, ANC visits, emergency health communication and improving overall awareness [12,13].

Most existing literature primarily focuses on factors influencing the frequency of ANC visits, while only a limited number of studies, particularly those involving Sub-Saharan African data, have examined the factors that affect the timing of ANC visits [14–18]. This underscores a gap in the literature regarding the relationship between late initiation of ANC and subsequent low ANC attendance. The objective of this study was to address this gap by analyzing the factors associated with delayed and insufficient ANC visits in Bangladesh and examining the impact of late ANC initiation on the overall number of ANC visits, using nationally representative data from 2007 and 2017–18. We specifically wanted to assess the change in ANC service structure in Bangladesh across one decade and thus we chose to use 2007 and 2017–18 survey datasets.

### Theoretical framework

This study investigated the factors associated with the low number of ANC visits in Bangladesh through the lens of healthcare access theory. We adopted a modified version of the theoretical framework used by Afiaz & Biswas (2021) [19], who utilized Saurman's (2016) [20] extended version of the original healthcare access theory proposed by Penchansky & Thomas (1981) [21]. Saurman's (2016) [20] extended version included an additional dimension called 'awareness' on top of the existing five dimensions of acceptability, availability, affordability, accessibility, and adequacy prescribed by Penchansky & Thomas (1981) [21]. While our analysis explores aspects such as accessibility, acceptability, affordability, adequacy, and awareness of ANC visits, data constraints limit the exploration of the dimension of availability (Fig 1). Our findings aim to inform targeted interventions aimed at optimizing timing and the number of ANC visits, incorporating both hardware and software strategies to address knowledge gaps and promote maternal and child health.

## Methods

### Data overview

The data used in this analysis comes from the Bangladesh Demographic and Health Survey (BDHS). BDHS is a part of the Demographic Health Survey program conducted in LMICs by USAID, collects accurate, nationally representative data on fertility, family planning, maternal and child health, HIV/AIDS, and nutrition [22]. We utilized the Bangladesh Demographic and Health Survey (BDHS) data sets from 2007 (n = 3050) and 2017–18 (n = 4544). Both of these two BDHS 2007 and 2017–18 surveys were chosen to allow for a comparative analysis over a decade, capturing potential changes in antenatal care (ANC) utilization patterns and associated sociodemographic factors over time. The sampling frame for both surveys consisted of a list of enumeration areas (EAs) derived from the census. Employing a two-stage stratified cluster sampling approach with probability proportional to size sampling method, the initial stage comprised selecting 361 EAs in 2007 and 675 EAs in 2017–18. Subsequently, a second stage involved employing systematic sampling method to select an average of 30 households from each EA. In the 2017–18 BDHS, data collection occurred in a total of 20,160 households and 10,819 households made up the sample of data for the 2007 BDHS.

### Outcome variables and covariates

We investigated two binary outcome variables pertaining to the ANC visits. The first outcome variable was for an indicator for whether the timing of the ANC visit was late (after 12 weeks). We will refer to this as late ANC for brevity from here on. The second outcome variable was the indicator for whether the number of ANC visits were 'less than four', which will

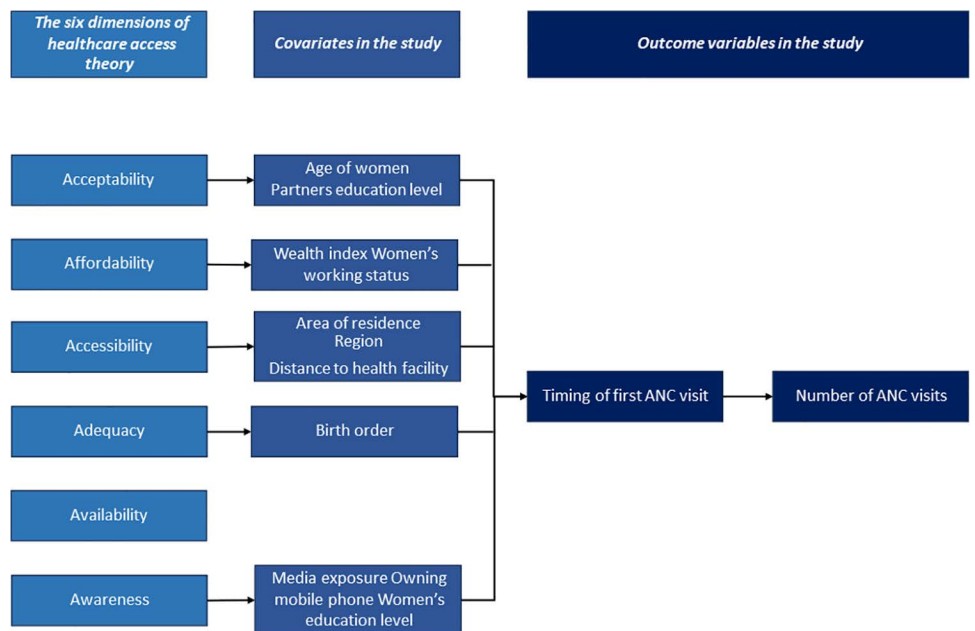

**Fig 1. Theorical framework for the current studies regarding timing and number of ANC visit.**

be referred to as low ANC from here on. While Bangladesh national guidelines define late ANC initiation as the first visit occurring after 16 weeks of gestation, this study adopted WHO's stricter >12-week threshold to align with global standards and enhance cross-LMIC comparability. For ANC frequency (i.e., Low ANC), we analyzed both WHO's updated 8-visit benchmark and Bangladesh's 4-visit guideline, prioritizing the latter in primary analyses to reflect local policy and the 2007 and 2017–18 BDHS datasets. Supplemental use of the WHO 8-visit threshold produced directionally consistent results but amplified the effect sizes (detailed in S1, S2, S4, and S5 Tables).

The covariates included in the study were as follows: area of residence, wealth index, partner's education level, women's education level, women's working status, media exposure, distance to health facility, birth order, division/region, women's age, and owning mobile phone. Note that the divisional structure differed between the 2007 and 2017–18 surveys due to administrative boundary changes in Bangladesh. The Dhaka division was divided into Mymensingh and Dhaka, while the Rajshahi division was divided into Rangpur and Rajshahi. Detailed information on this is provided in the supplementary document (S1 Text).

## Statistical analysis

Given that the study utilized large-scale survey data, missing cases were handled using listwise deletion, assuming that the data were missing at random to ensure unbiased estimates [23,24]. An initial summary of the data was provided by a descriptive statistics table, accompanied by a visualization of the spatial distribution of late ANC initiation and low ANC attendance through descriptive statistics and geographical mapping. To examine the factors associated with these outcomes, complex survey-weighted adjusted logistic regression models with a quasi-binomial family distribution were employed, enabling the analysis of binary outcomes while accounting for overdispersion. Multicollinearity was assessed using generalized variance inflation factor (GVIF) scores for each predictor.

We utilized classification tree analysis to investigate complex interactions between predictors and nonlinear relationships between predictor variables and the outcome measures [25]. It partitions data using hypothesis testing at each

node. The null hypothesis is $H_{0j} : D(\mathbf{Y}|X_j) = D(\mathbf{Y})$, which assesses whether the response variable $\mathbf{Y}$ is independent of each covariate $X_j$; j = 1, 2, …, m. The test statistics is defined as:

$$T_j(L_n, w) = \text{vec}\left(\sum_{i=1}^{n} w_i g_j(X_{ji}) h(Y_i, (Y_1, Y_2, \ldots, Y_n))'\right),$$

where $g_j(X_{ji})$ is a transformation of the predictor, $h(Y_i, (Y_1, Y_2, \ldots, Y_n))'$ represents the influence function of the response, and $w_i$ denotes case weight. The test statistic is standardized using its conditional expectation ($\mu_j$) and covariance ($\Sigma_j$) under the null hypothesis. If $H_{0j}$ is rejected, the algorithm selects the covariate with the strongest association (smallest p-value). In the next step it determines the optimal split by maximizing a discrepancy measure:

$$A^* = arg \ \max_A \ c(T_A, \mu_A, \Sigma_A)$$

The recursion halts when no significant association is found at a predefined significance level ($\alpha$).

The results of the classification tree analysis provide useful insights. The tree structure visually interprets the importance of covariates in splitting data. The significance of splits is determined by p-values, with corrections like Bonferroni adjustments applied to handle multiple comparisons. Terminal nodes provide a summary of the response variable within the distinct partitions.

Statistical software R (version 4.3.1) [26] was used for performing all statistical analyses, including creating the geographical maps in this study. The analysis utilized several R packages: tidyverse [27], survey [28], ggplot2 [29], partykit [25] and gridExtra [30]. Besides, from an openly licensed database called geoBoundaries Global Administrative Database, we obtained the administrative boundary shapefile of Bangladesh [31].

### Ethical Considerations

This work does not include any studies with human participants conducted by the authors. Therefore, no ethical approval was needed for our study as we are using secondary data sets provided by the Demographic Health Survey (DHS). The survey protocol for the Bangladesh DHS (BDHS) datasets utilized in this study received approval by institutional review boards (IRBs) at ICF and the Bangladesh Medical Research Council (BMRC). Written informed consent was obtained from each participant before conducting the interviews. Participants who did not provide consent were excluded from the survey. Participants were made aware that their participation was entirely voluntary. They were assured that their responses would be kept confidential and anonymous. Additionally, they were informed of their right to skip any questions they preferred not to answer and could withdraw from the interview at any point. All data were fully anonymized before the authors were given access. The datasets are available on request on the Demographic and Health Survey (DHS) website free of cost for research purposes (https://dhsprogram.com/).

### Results

#### Descriptive analysis

The distribution of late initiation of ANC and less than four ANC visits by sociodemographic factors is shown in Table 1. In both 2007 and 2017–18, approximately 60% of women had a delayed first ANC visit. Meanwhile, low ANC was 65.8% in 2007 and decreased to 48.8% in 2017, which shows that over the ten years, late ANC remained same, but low ANC decreased by 17%. In 2007 and 2017–18, respectively, 80.2% and 64.8% of women who started their ANC visit late also had less than four ANC visits, whereas women who were not late, respectively, 55.4% and 74.9%, had over four ANC visits. In both 2007 and 2017–18, rural women had higher late ANC visits (15.2% and 13.3%, respectively) and also reported higher low ANC visits (22.2% and 15.2%, respectively) than women in urban areas. The poorest, less educated, and non-media-exposed women had more late ANC and low ANC visits compared to the other categories despite the survey years. In 2017–18 data, women who

**Table 1. The frequency distribution of late ANC and low ANC (< 4) visits by sociodemographic factors.**

| Variables | BDHS 2007 | | | BDHS 2017–18 | | |
|---|---|---|---|---|---|---|
| | Total sample, N = 2,949 (%) | N (%) of women with Late ANC visits | N (%) of women with Low (< 4) ANC visits | Total sample, N = 4,588 (%) | N (%) of women with Late ANC visits | N (%) of women with Low (< 4) ANC visits |
| **Area of residence** | | | | | | |
| Urban | 783 (26.5) | 379 (48.4) | 387 (49.5) | 1,266 (27.6) | 634 (50.0) | 479 (37.8) |
| Rural | 2,166 (73.5) | 1,377 (63.6) | 1,553 (71.7) | 3,322 (72.4) | 2,104 (63.3) | 1,761 (53.0) |
| **Wealth index** | | | | | | |
| Poorest | 446 (15.1) | 325 (72.9) | 357 (80.2) | 848 (18.5) | 620 (73.2) | 531 (62.6) |
| Poorer | 495 (16.8) | 339 (68.5) | 393 (79.4) | 912 (19.9) | 605 (66.3) | 539 (59.0) |
| Middle | 549 (18.6) | 354 (64.6) | 410 (74.7) | 894 (19.5) | 561 (62.8) | 459 (51.4) |
| Richer | 688 (23.3) | 397 (57.8) | 433 (62.9) | 970 (21.2) | 582 (60.0) | 447 (46.0) |
| Richest | 772 (26.2) | 340 (44.1) | 347 (45.0) | 964 (21.0) | 369 (38.2) | 265 (27.4) |
| **Region** | | | | | | |
| Dhaka | 882 (29.9) | 505 (57.3) | 575 (65.3) | 1,185 (25.8) | 595 (50.2) | 531 (44.8) |
| Barishal | 164 (5.6) | 95 (57.6) | 113 (68.6) | 241 (5.3) | 155 (64.2) | 135 (55.8) |
| Chattogram | 617 (20.9) | 343 (55.5) | 416 (67.4) | 966 (21.1) | 633 (65.5) | 556 (57.5) |
| Khulna | 356 (12.1) | 214 (60.1) | 230 (64.7) | 439 (9.6) | 263 (60.0) | 177 (40.3) |
| Mymensingh | | | | 381 (8.3) | 228 (59.8) | 189 (49.6) |
| Rajshahi | 721 (24.4) | 479 (66.4) | 446 (61.8) | 550 (12.0) | 366 (66.4) | 273 (49.5) |
| Rangpur | | | | 502 (10.9) | 322 (64.0) | 187 (37.2) |
| Sylhet | 209 (7.1) | 121 (58.1) | 161 (77.0) | 323 (7.0) | 176 (54.4) | 192 (59.4) |
| **Women's age (Mean [SD])** | 24.9 (5.8) | 24.5 (5.8) | 24.8 (5.9) | 24.8 (5.5) | 24.7 (5.6) | 24.8 (5.6) |
| **Women's education level** | | | | | | |
| No education | 474 (16.1) | 311 (65.7) | 389 (82.0) | 229 (5.0) | 166 (72.7) | 166 (72.7) |
| Primary | 828 (28.1) | 541 (65.3) | 636 (76.8) | 1,198 (26.1) | 841 (70.2) | 731 (61.0) |
| Secondary | 1,340 (45.4) | 787 (58.8) | 816 (60.9) | 2,314 (50.4) | 1,404 (60.7) | 1,087 (47.0) |
| Higher | 307 (10.4) | 117 (38.0) | 100 (32.4) | 847 (18.5) | 326 (38.5) | 256 (30.2) |
| **Women's employment status** | | | | | | |
| Not Working | 2,193 (74.4) | 1,293 (58.9) | 1,423 (64.9) | 2,919 (63.6) | 1,653 (56.6) | 1,419 (48.6) |
| Working | 756 (25.6) | 464 (61.3) | 517 (68.4) | 1,669 (36.4) | 1,084 (64.9) | 821 (49.2) |
| **Partner's education level** | | | | | | |
| No education | 734 (24.9) | 516 (70.3) | 588 (80.1) | 576 (12.5) | 421 (73.2) | 369 (64.0) |
| Primary | 775 (26.3) | 483 (62.3) | 570 (73.6) | 1,488 (32.4) | 1,025 (68.9) | 864 (58.1) |
| Secondary | 964 (32.7) | 562 (58.3) | 596 (61.8) | 1,617 (35.3) | 944 (58.4) | 761 (47.0) |
| Higher | 476 (16.1) | 195 (41.0) | 187 (39.2) | 907 (19.8) | 347 (38.2) | 247 (27.2) |
| **Media exposure** | | | | | | |
| No | 795 (27.0) | 532 (66.9) | 619 (77.8) | 1,458 (31.8) | 1,021 (70.0) | 909 (62.4) |
| Yes | 2,154 (73.0) | 1,225 (56.9) | 1,322 (61.4) | 3,130 (68.2) | 1,717 (54.8) | 1,331 (42.5) |
| **Birth order** | | | | | | |
| 1 | 1,162 (39.4) | 669 (57.6) | 701 (60.4) | 1,815 (39.6) | 1,007 (55.5) | 795 (43.8) |
| 2-3 | 1,298 (44.0) | 795 (61.3) | 849 (65.4) | 2,276 (49.6) | 1,373 (60.3) | 1,109 (48.7) |
| 4+ | 489 (16.6) | 292 (59.7) | 390 (79.8) | 498 (10.8) | 358 (71.9) | 336 (67.5) |

*(Continued)*

**Table 1.** (Continued)

| Variables | BDHS 2007 | | | BDHS 2017–18 | | |
|---|---|---|---|---|---|---|
| | Total sample, N = 2,949 (%) | N (%) of women with Late ANC visits | N (%) of women with Low (< 4) ANC visits | Total sample, N = 4,588 (%) | N (%) of women with Late ANC visits | N (%) of women with Low (< 4) ANC visits |
| **Distance to health facility** | | | | | | |
| Not a big problem | | | | 2,758 (60.1) | 1,563 (56.7) | 1,230 (44.6) |
| Big problem | | | | 1,830 (39.9) | 1,174 (64.14) | 1,010 (55.2) |
| **Owning mobile phone** | | | | | | |
| No | | | | 1,670 (36.4) | 1,136 (68.0) | 968 (57.9) |
| Yes | | | | 2,918 (63.6) | 1,602 (54.9) | 1,272 (43.6) |
| **Timing of first ANC visit** | | | | | | |
| Not late | 1,193 (40.4) | | 532 (44.6) | 1,851 (40.3) | | 465 (25.1) |
| Late | 1,756 (59.6) | | 1,409 (80.2) | 2,737 (59.7) | | 1,775 (64.8) |
| **Number of ANC visit** | | | | | | |
| ≥4 ANC | 1009 (34.2) | | | 2,348 (51.2) | | |
| Low (<4) ANC | 1,940 (65.8) | | | 2,240 (48.8) | | |

reported that the distance to health facilities was a problem had, respectively, 7.4% and 10.6% higher late ANC and low ANC visits compared to the women who had no issue with the distance to the health facility. Women whose most recent birth was of order four or higher reported greater percentage of late and low ANC visits compared to the women giving birth to a first, second, or third child. On the other hand, women who owned mobile phones reported 13.1% and 14.3% lower percentage of late and low ANC visits compared with women who did not own one. Women with partners with higher education also reported a lower percentage of late and low ANC visitscompared to those with primary or no education.

The distribution of late initiation of ANC and less than eight ANC visits by sociodemographic factors, as well as the frequency distribution of low ANC visits (<8) by late ANC is provided supplementary document (S1 and S2 Tables).

## Logistic regression models

The results of binary logistic regression model adjusted for sociodemographic factors with timing of first ANC visit and number of ANC visits (low (< 4) ANC visits) as outcome is shown in Table 2. Among urban and rural mothers, in 2007, rural mothers were associated with 26% higher odds (AOR: 1.26 [95% CI: 1.03, 1.53]; p-value = 0.023) of having a late ANC visit compared to urban mothers. In terms of the number of ANC visits, rural mothers were associated with 30% (AOR: 1.30 [1.08, 1.55]; p-value = 0.005) and 71% (AOR: 1.71 [1.38, 2.11]; p-value < 0.001) higher odds of having low ANC visits, respectively, in 2017–18 and 2007, compared to urban mothers. For each one-year difference in women's age, older women were associated with 2% lower (AOR: 0.98 [95% CI: 0.96, 1.00]; p-value = 0.036) and 3% lower odds (AOR: 0.97 [95% CI: 0.95, 0.99]; p-value = 0.003) of having a late ANC visit in 2017–18 and 2007, respectively. Similarly, the odds of having a low number of ANC visits were found lower by 3% for each one year of age in both 2017–18 (AOR: 0.97 [95% CI: 0.95, 0.99]; p-value < 0.001) and 2007 (AOR: 0.97 [95% CI: 0.95, 0.99]; p-value = 0.011). Despite the survey year, the poorest, least educated, and, in 2017–18 non-media-exposed women were associated with higher odds of having late and low number of ANC visits compared to the reference categories. Besides, for 2017–18, those for whom distance to health facilities was a problem were associated with 20% higher odds (AOR: 1.20 [1.03, 1.38]; p-value = 0.016) of having low ANC visits compared to those who did not have any problem with the distance. In the case of women

**Table 2. Binary logistic regression model adjusted for sociodemographic factors with timing of first ANC visit and number of ANC visits (low (< 4) ANC visits) as outcome.**

| Variables | Late ANC visits | | | | | | Low (< 4) ANC visits | | | | | |
|---|---|---|---|---|---|---|---|---|---|---|---|---|
| | BDHS 2007 | | | BDHS 2017–18 | | | BDHS 2007 | | | BDHS 2017–18 | | |
| | AOR | 95% CI | p-value | AOR | 95% CI | p-value | AOR | 95% CI | p-value | AOR | 95% CI | p-value |
| **Area of residence** | | | | | | | | | | | | |
| Urban (ref.) | — | — | | — | — | | — | — | | — | — | |
| Rural | 1.26 | 1.03, 1.53 | **0.023** | 1.14 | 0.97, 1.35 | 0.122 | 1.71 | 1.38, 2.11 | **<0.001** | 1.30 | 1.08, 1.55 | **0.005** |
| **Wealth index** | | | | | | | | | | | | |
| Poorest (ref.) | — | — | | — | — | | — | — | | — | — | |
| Poorer | 0.85 | 0.60, 1.20 | 0.358 | 0.88 | 0.69, 1.12 | 0.296 | 1.04 | 0.69, 1.56 | 0.842 | 1.06 | 0.83, 1.34 | 0.658 |
| Middle | 0.78 | 0.56, 1.09 | 0.140 | 0.91 | 0.71, 1.18 | 0.489 | 0.93 | 0.63, 1.39 | 0.739 | 0.86 | 0.68, 1.10 | 0.236 |
| Richer | 0.66 | 0.47, 0.92 | **0.014** | 1.00 | 0.76, 1.31 | 0.981 | 0.68 | 0.46, 1.01 | 0.058 | 0.79 | 0.61, 1.02 | 0.074 |
| Richest | 0.49 | 0.34, 0.72 | **<0.001** | 0.58 | 0.43, 0.79 | **<0.001** | 0.49 | 0.31, 0.76 | **0.002** | 0.46 | 0.34, 0.63 | **<0.001** |
| **Region** | | | | | | | | | | | | |
| Dhaka (ref.) | — | — | | — | — | | — | — | | — | — | |
| Barishal | 0.83 | 0.62, 1.13 | 0.238 | 1.40 | 1.07, 1.82 | **0.013** | 0.87 | 0.62, 1.23 | 0.422 | 1.05 | 0.80, 1.36 | 0.742 |
| Chattogram | 0.93 | 0.73, 1.19 | 0.556 | 1.73 | 1.38, 2.17 | **<0.001** | 1.09 | 0.84, 1.43 | 0.516 | 1.43 | 1.14, 1.81 | **0.002** |
| Khulna | 1.03 | 0.77, 1.37 | 0.845 | 1.29 | 0.99, 1.68 | 0.063 | 0.88 | 0.65, 1.19 | 0.410 | 0.66 | 0.50, 0.88 | **0.004** |
| Mymensingh | | | | 1.09 | 0.82, 1.44 | 0.563 | | | | 0.77 | 0.58, 1.02 | 0.069 |
| Rajshahi | 1.25 | 0.96, 1.63 | 0.093 | 1.63 | 1.24, 2.14 | **<0.001** | 0.65 | 0.48, 0.87 | **0.004** | 0.92 | 0.71, 1.20 | 0.542 |
| Rangpur | | | | 1.32 | 0.99, 1.74 | 0.055 | | | | 0.44 | 0.33, 0.58 | **<0.001** |
| Sylhet | 0.89 | 0.63, 1.25 | 0.508 | 0.79 | 0.61, 1.02 | 0.071 | 1.28 | 0.88, 1.88 | 0.199 | 1.09 | 0.84, 1.41 | 0.510 |
| **Women's age** | 0.97 | 0.95, 0.99 | **0.003** | 0.98 | 0.96, 1.00 | **0.036** | 0.97 | 0.95, 0.99 | **0.011** | 0.97 | 0.95, 0.99 | **<0.001** |
| **Women's education level** | | | | | | | | | | | | |
| No education (ref.) | — | — | | — | — | | — | — | | — | — | |
| Primary | 1.04 | 0.78, 1.40 | 0.787 | 0.93 | 0.65, 1.34 | 0.702 | 0.89 | 0.62, 1.28 | 0.517 | 0.62 | 0.42, 0.89 | **0.010** |
| Secondary | 0.98 | 0.71, 1.36 | 0.911 | 0.76 | 0.53, 1.09 | 0.130 | 0.57 | 0.39, 0.83 | **0.004** | 0.48 | 0.33, 0.70 | **<0.001** |
| Higher | 0.77 | 0.49, 1.20 | 0.250 | 0.52 | 0.35, 0.78 | **0.002** | 0.36 | 0.22, 0.59 | **<0.001** | 0.45 | 0.29, 0.70 | **<0.001** |
| **Women's employment status** | | | | | | | | | | | | |
| Not Working (ref.) | — | — | | — | — | | — | — | | — | — | |
| Working | 0.93 | 0.76, 1.15 | 0.524 | 1.08 | 0.94, 1.26 | 0.281 | 1.03 | 0.83, 1.29 | 0.768 | 0.85 | 0.73, 0.99 | **0.039** |
| **Partner's education level** | | | | | | | | | | | | |
| No education (ref.) | — | — | | — | — | | — | — | | — | — | |
| Primary | 0.75 | 0.58, 0.97 | **0.030** | 0.92 | 0.71, 1.20 | 0.558 | 0.88 | 0.65, 1.18 | 0.385 | 0.99 | 0.78, 1.25 | 0.944 |
| Secondary | 0.79 | 0.60, 1.04 | 0.098 | 0.73 | 0.56, 0.96 | **0.023** | 0.76 | 0.56, 1.03 | 0.078 | 0.82 | 0.64, 1.05 | 0.120 |
| Higher | 0.52 | 0.36, 0.76 | **<0.001** | 0.49 | 0.36, 0.68 | **<0.001** | 0.53 | 0.37, 0.77 | **<0.001** | 0.53 | 0.39, 0.73 | **<0.001** |
| **Media exposure** | | | | | | | | | | | | |
| No (ref.) | — | — | | — | — | | — | — | | — | — | |
| Yes | 0.98 | 0.77, 1.23 | 0.848 | 0.79 | 0.67, 0.94 | **0.008** | 0.98 | 0.75, 1.28 | 0.885 | 0.72 | 0.60, 0.85 | **<0.001** |
| **Birth order** | | | | | | | | | | | | |
| 1 (ref.) | — | — | | — | — | | — | — | | — | — | |
| 2-3 | 1.29 | 1.04, 1.60 | **0.022** | 1.17 | 0.97, 1.42 | 0.097 | 1.25 | 0.99, 1.57 | 0.061 | 1.32 | 1.10, 1.59 | **0.003** |
| 4+ | 1.19 | 0.82, 1.72 | 0.357 | 1.55 | 1.08, 2.24 | **0.018** | 1.92 | 1.26, 2.93 | **0.002** | 2.31 | 1.64, 3.24 | **<0.001** |
| **Distance to health facility** | | | | | | | | | | | | |
| Not a big problem (ref.) | | | | — | — | | | | | — | — | |
| Big problem | | | | 1.1 | 0.94, 1.27 | **0.233** | | | | 1.20 | 1.03, 1.38 | **0.016** |

*(Continued)*

**Table 2.** (Continued)

| Variables | Late ANC visits | | | | | | Low (< 4) ANC visits | | | | | |
|---|---|---|---|---|---|---|---|---|---|---|---|---|
| | BDHS 2007 | | | BDHS 2017–18 | | | BDHS 2007 | | | BDHS 2017–18 | | |
| | AOR | 95% CI | p-value | AOR | 95% CI | p-value | AOR | 95% CI | p-value | AOR | 95% CI | p-value |
| **Owning mobile phone** | | | | | | | | | | | | |
| No (ref.) | | | | — | — | | | | | — | — | |
| Yes | | | | 0.84 | 0.72, 0.97 | **0.021** | | | | 0.77 | 0.66, 0.89 | **<0.001** |

AOR = Adjusted Odds Ratio, CI = Confidence Interval

owning mobile phones, there were associated with, respectively, 16% (AOR: 0.84 [0.72, 0.97]; p-value = 0.021) and 23% (AOR: 0.77 [0.66, 0.89]; p-value < 0.001) lower odds of having late and low ANC visits compared to women without a mobile phone. Irrespective of the survey year, in the case of the highest birth order, mothers were associated with higher odds of having late and low ANC visits compared to if they were giving birth to a first, second, or third child. Women with higher-educated partners were also associated with lower odds of having late and low ANC coverage compared to those having partners with primary or no education.

In 2017–18, women who reported late ANC initiation were associated with 4.68 times (AOR: 4.68 [95% CI: 4.00, 5.48]; p-value < 0.001) the odds of having less than four ANC visits compared to those who were not late after adjusting for the others covariate, which remained unchanged in 2007 (AOR: 4.60 [95% CI: 3.69, 5.73]; p-value < 0.001), shown in Table 3 and detailed in S3 Table.

The supplementary document (S4 and S5 Tables) provides binary logistic regression models adjusted for sociodemographic factors and the timing of the first ANC visit, with number of ANC visits as outcome categorized as fewer than eight ANC visits or more than eight ANC visits.

## Spatial mapping

The map presented in Fig 2 illustrates the regional distribution of late and low (< 4) ANC visits in Bangladesh. In 2007, Rajshahi and Rangpur regions showed the highest percentage of late ANC visits at 66.4%, while Chattogram region had the lowest at 55.5%. Conversely, in 2017–18, Rajshahi region maintained the highest percentage of late ANC visits at 66.4%, with Dhaka region recording the lowest at 50.2%. This indicates notable changes in the regional distribution of late ANC visits over the decade, with nearly half of the regions showing improvement while others worsening.

Regarding low ANC visits, in 2007, the Sylhet division had the highest percentage at 77%, while Rajshahi and Rangpur regions had the lowest at 61.8%. By 2017–18, Sylhet region still had the highest percentage at 59.4%, whereas Rangpur

**Table 3. Binary logistic regression model adjusted for sociodemographic factors and timing of first ANC visit with number of ANC visits (low (< 4) ANC visits) as outcome.**

| Characteristic | BDHS 2007 | | | BDHS 2017–18 | | |
|---|---|---|---|---|---|---|
| | AOR | 95% CI | p-value | AOR | 95% CI | p-value |
| **Timing of first ANC visit** | | | | | | |
| Not late (ref.) | — | — | | — | — | |
| Late | 4.60 | 3.69, 5.73 | **<0.001** | 4.68 | 4.00, 5.48 | **<0.001** |

AOR, adjusted odds ratio; CI, confidence interval.

Adjusted for area of residence, wealth index, region, women's age, women's education level, women's employment status, partner's education level, media exposure, birth order, distance to health facility, and owning mobile phone.

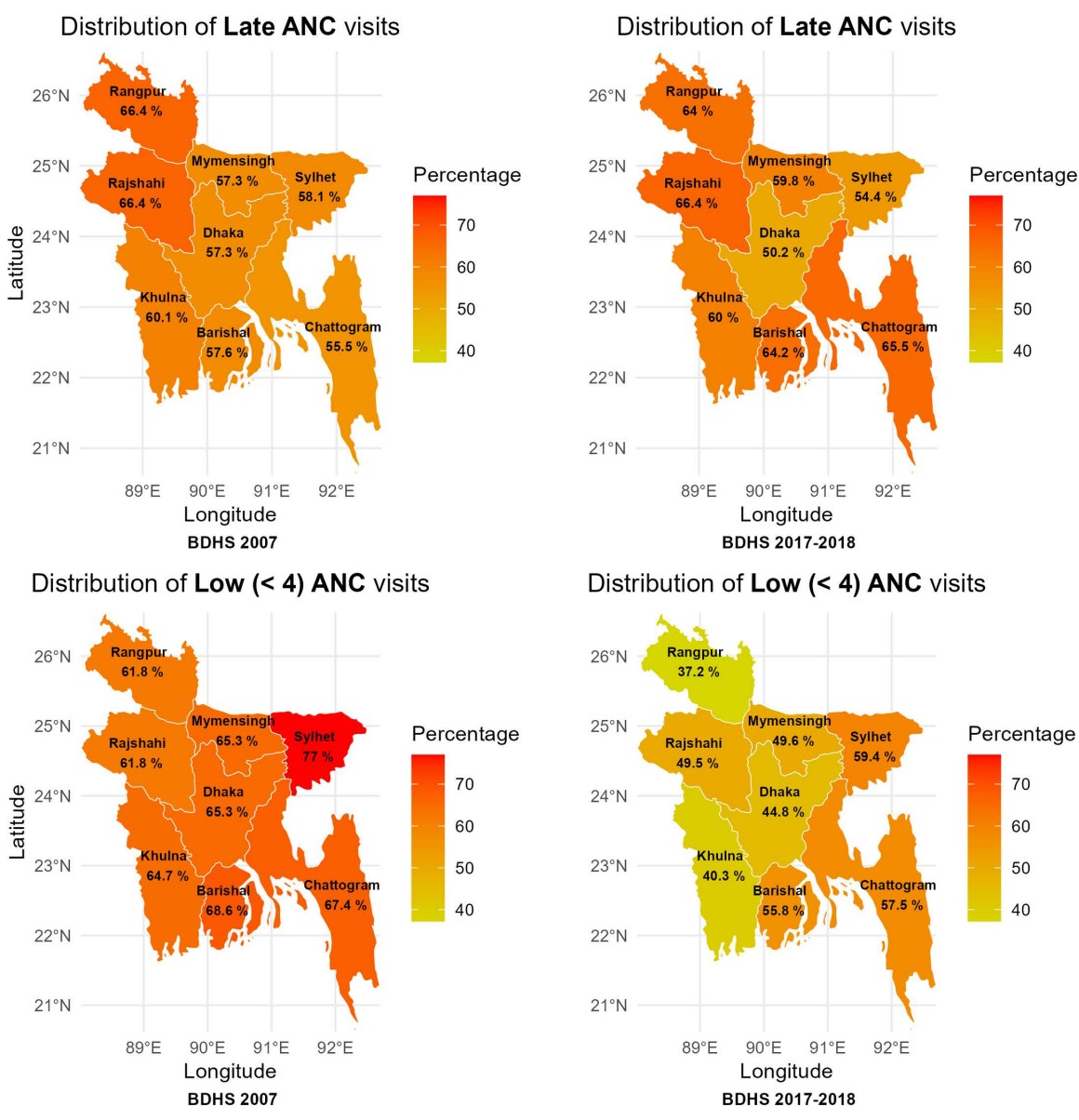

**Fig 2. Region-wise percentage distribution of late and low (< 4) ANC visit in Bangladesh.**

region had the lowest at 37.2%. This suggests improvement across almost all regions in terms of the number of ANC visits over the decade.

## Classification trees

According to the classification trees (Fig 3), in both 2007 and 2017–18, the most important predictor of the low number of ANC visits was the timing of the first ANC visit appearing at the top of the tree. Administrative divisions (region) also remained very important across the two time points. Notice that most of the nodes on the right (which corresponded to late

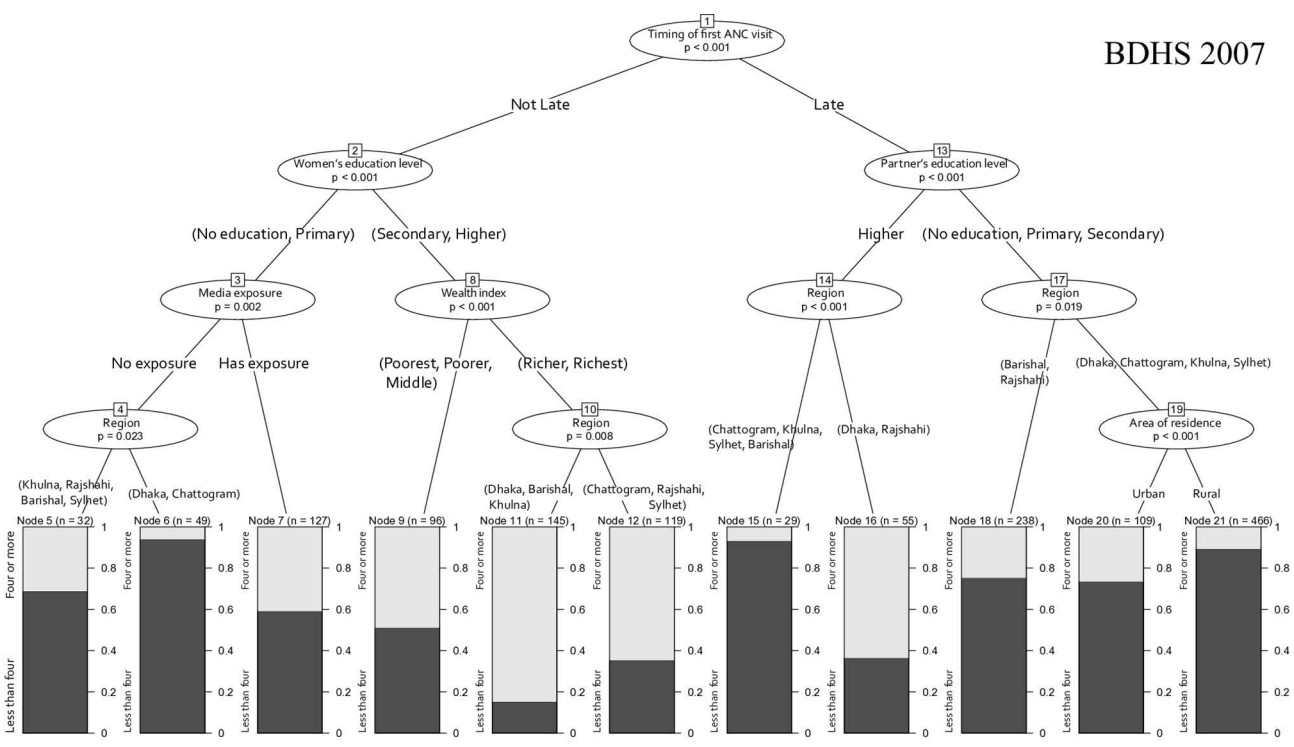

BDHS 2007

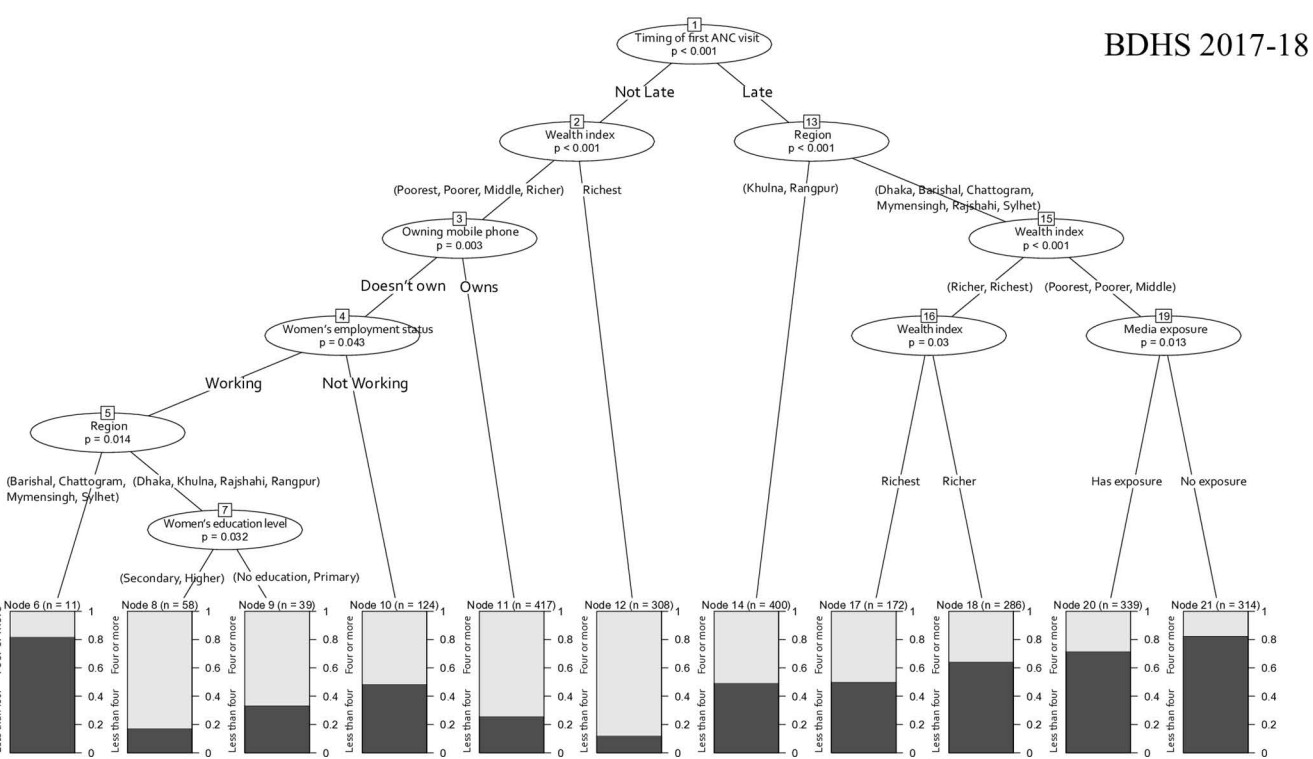

BDHS 2017-18

**Fig 3. Classification tree illustrating the important predictors of low ANC visits in Bangladesh for 2007 and 2017-18.**

ANC initiation) had a higher prevalence of low ANC (< 4), which highlighted the importance of early initiation in ensuring adequate ANC coverage. Wealth index, mobile ownership, media exposure also became prominent for the most recent survey data and coincided with our earlier findings from the regression models.

## Model validation

The supplementary document (S6, S7, S8, S9, and S10 Tables) includes generalized variance inflation factors (GVIF) analysis for binary logistic regression models to assess multicollinearity. All the squared adjusted GVIF values are below five, indicating that our regression models do not have multicollinearity issues [32].

## Discussion

Our study explored the association between late initiation of ANC and subsequent low ANC attendance, highlighting the critical importance of early ANC visits and their relationship with achieving the recommended number of ANC visits. We identified sociodemographic factors associated with both timing and total number of ANC visits. The findings revealed that, in both 2007 and 2017–18, approximately 60% of women delayed their first ANC visit, with a slight increase observed over the decade. While crude prevalence estimates for low ANC visits decreased by 17%, nearly half of the women (approximately 50%) still did not meet the minimum number of ANC visits recommended by the WHO FANC model and national guidelines in Bangladesh. Importantly, the factors associated with both late and insufficient ANC visits remained largely unchanged over the 10-year period. Our results underscore that, despite governmental efforts to improve maternal healthcare, sociodemographic factors such as education, socioeconomic status, maternal age, and birth order have continued to influence the timing and frequency of ANC visits throughout this period. Furthermore, Bangladesh national guidelines need to revisit the recommended number and timing of ANC visits to align with WHO standards. This would enhance the understanding, help in cross-country comparisons and adapting global strategies to improve maternal health outcomes.

Our analysis revealed a strong association between the timing of the first ANC visit and the subsequent number of ANC visits, as evidenced by the regression model results. The classification tree analysis further confirmed that the timing of the first ANC visit was the most significant predictor of number of ANC visits, with consistent findings across both the 2007 and 2017–18 surveys. These results are particularly pertinent to LMICs such as Bangladesh, where traditional societal norms and cultural beliefs exert a substantial influence on healthcare-seeking behaviors [33,34]. In Bangladeshi rural areas, pregnancy is often regarded as a natural process that does not require early medical intervention unless complications arise, leading to delayed ANC initiation [35]. Additionally, decision-making regarding maternal health is frequently influenced by husbands or elder family members [33,35], who may not prioritize early ANC visits due to traditional views or a lack of awareness. Furthermore, rural women, who are the most susceptible to being influenced by traditional norms, also showed higher odds of late and low ANC visits compared to urban mothers, irrespective of the survey year, a trend supported by studies in Bangladesh [36] and Vietnam [37]. Additionally, women facing challenges with distance to healthcare facilities showed higher odds of having a low number of ANC visits. Studies conducted in Burkina Faso [38], and Benin [39] further support to this finding.

Despite increases in women's education and employment, many women in Bangladesh continue to face significant barriers in accessing healthcare and ANC. Issues including cultural norms, poor healthcare facilities, and financial challenges make it difficult for women in accessing healthcare facilities [40]. Women from less affluent households had higher odds of having late initiation and a low number of ANC visits compared to the others, indicating socioeconomic inequalities in ANC coverage in Bangladesh. Financial barriers significantly impact access to maternal healthcare services, as lower-income families may prioritize immediate basic needs over healthcare expenses, leading to delayed or irregular ANC visits [41,42]. Evidence from studies conducted in Ethiopia [43,44], and Cameroon [17] also supports this conclusion. While most regions have shown improvement over the decade, regional disparities still exist, highlighting the ongoing need to address regional inequalities in maternal healthcare access in Bangladesh, as evident from recent studies [45]. Regional variations in ANC utilization also

evident in other LMICs like Ethiopia [46] and Indonesia [47]. Women with higher education levels had reduced odds of experiencing late initiation and insufficient ANC visits compared to those with lower levels of education which underscored the crucial role of mother's education in promoting timely and adequate ANC attendance [41,48,49]. This assertion finds additional backing from research conducted in other LMICs including Nigeria [50,51], Ethiopia [15,46], and Indonesia [47].

Our findings underscored the importance of accessing information and healthcare facilities in ensuring timely and adequate ANC visits. Access to information through media and mobile phones enhances awareness and understanding of the importance of early and regular ANC visits, ultimately improving maternal healthcare utilization [41,48], highlighting the importance of media exposure, internet use, and mobile phones on reproductive healthcare services. A recent study in Bangladesh also highlights the crucial role of internet use, and mobile phone on reproductive health service [52]. Maternal age also has been identified as a significant factor associated with both the timing and number of ANC visits. Previous study conducted in Ethiopia [53], Ghana [54], and Gambia [55] also found maternal age associated with timing and number of ANC visit.

We found that women who gave birth to three or more children previously had increased odds of experiencing late initiation and <4 ANC visits compared to those who had less than two children. This phenomenon can be attributed to several factors. Firstly, mothers with multiple children often face increased household responsibilities [56] and time constraints, which can limit their ability to prioritize and attend regular ANC visits. Additionally, with subsequent births, mothers may consider themselves more experienced and might underestimate the importance of early and frequent ANC visits [57]. This self-satisfaction can lead to reduced utilization of maternal health services [58]. This assertion finds additional backing from research conducted in Ethiopia [44], and Cameroon [17]. Although the study is cross-sectional, valuable policy insights can be drawn from the findings. Strengthening community-based interventions, such as targeted health education campaigns and home visits by community health workers, can improve awareness and encourage early ANC initiation. Integrating ANC services within existing maternal and child health programs and ensuring financial and geographic accessibility, particularly for socioeconomically disadvantaged groups, may reduce barriers to care. Recognizing women's right to access healthcare during pregnancy and childbirth is essential to reducing the risk of complications and ensuring equitable maternal health outcomes. Policymakers should consider revising national ANC guidelines to emphasize early initiation and continuity of care at the global standard, ensuring they are effectively disseminated and implemented across all healthcare facilities.

The primary strength of our study was that we analyzed nationally representative data spanning over a decade, from 2007 to 2017–18. This extensive timeframe enabled us to explore the persistent factors contributing to late initiation and insufficient ANC visits among women in Bangladesh, which remain unresolved and urgently require attention. Notably, this is the first study of its kind to explore this relationship, utilizing multiple large, nationally representative datasets from Bangladesh Demographic and Health Survey (BDHS) pertaining to Bangladesh. Therefore, it paves the way for further studies in other LMICs in the future. However, several limitations should be noted. First, our study is observational and based on repeated surveys of different, yet representative, cohorts, which limits causal interpretation. Additionally, relying on self-reported data may introduce recall biases or social desirability biases. Some participants may not accurately recall the date of first ANC visit or may over or underreport the number of ANC visits. This recall bias regarding the timing and frequency of ANC visits in self-reported data could affect the validity of the findings. Future studies with health linkage or health insurance data could validate self-reported data with medical records or conducting longitudinal data collection to prevent recall bias. Employing a longitudinal design could better assess causal relationships and evaluate potential intervention strategies based on our findings. Finally, future research can consider applying models such as Spatial Lag Models to address spatial autocorrelation of ANC services in Bangladesh.

## Conclusion

Our study provides valuable insights into ANC utilization among women in Bangladesh from 2007 to 2017–18. We found that the timing of the first ANC visit was strongly associated with whether women received ≥ 4 ANC visits throughout their

pregnancy, underscoring the critical importance of early engagement with healthcare services. Despite ongoing efforts to promote maternal health, persistent demographic, socioeconomic, and regional disparities in ANC coverage remain. Women from rural areas, in particular, face barriers such as limited education, financial constraints, and cultural influences, which hinder their access to timely and ≥ 4 ANC visits compared to their urban counterparts. Addressing these disparities requires targeted policies that focus on enhancing maternal education, improving healthcare infrastructure, and promoting community-based initiatives that challenge traditional norms and raise awareness about the benefits of timely ANC visits.

## Supporting information

**S1 Table. The frequency distribution of late ANC and low (< 8) ANC visits by sociodemographic factors.**
(DOCX)

**S2 Table. The frequency distribution of low (<8) ANC visits by timing of first ANC visits.**
(DOCX)

**S3 Table. Binary logistic regression model adjusted for sociodemographic factors and timing of first ANC visit with number of ANC visits (low (<4) ANC visits) as outcome.**
(DOCX)

**S4 Table. Binary logistic regression model adjusted for sociodemographic factors with timing of first ANC visit and number of ANC visits (low (<8) ANC visits) as outcome.**
(DOCX)

**S5 Table. Binary logistic regression model adjusted for sociodemographic factors and timing of first ANC visit with number of ANC visits (low (<8) ANC visits) as outcome.**
(DOCX)

**S6 Table. GVIF for binary logistic regression model adjusted for sociodemographic factors with timing of first ANC visit as outcome.**
(DOCX)

**S7 Table. GVIF for binary logistic regression model adjusted for sociodemographic factors with number of ANC visits (low (<4) ANC visits) as outcome.**
(DOCX)

**S8 Table. GVIF for binary logistic regression model adjusted for sociodemographic factors and timing of first ANC visit with number of ANC visits (low (<4) ANC visits) as outcome.**
(DOCX)

**S9 Table. GVIF for binary logistic regression model adjusted for sociodemographic factors number of ANC visits (low (<8) ANC visits) as outcome.**
(DOCX)

**S10 Table. GVIF for binary logistic regression model adjusted for sociodemographic factors and timing of first ANC visit with number of ANC visits (low (<8) ANC visits) as outcome.**
(DOCX)

**S1 Text. Outcome variables and covariates.**
(DOCX)

## Acknowledgements

We would like to thank The Demographic and Health Surveys (DHS) Program for providing access to the datasets.

## Author contributions

**Conceptualization:** Md. Hasibul Islam Jitu, Awan Afiaz, Raaj Kishore Biswas.

**Data curation:** Md. Hasibul Islam Jitu.

**Formal analysis:** Md. Hasibul Islam Jitu.

**Methodology:** Md. Hasibul Islam Jitu, Awan Afiaz, Raaj Kishore Biswas.

**Supervision:** Awan Afiaz, Raaj Kishore Biswas.

**Visualization:** Md. Hasibul Islam Jitu, Awan Afiaz.

**Writing – original draft:** Md. Hasibul Islam Jitu, Awan Afiaz.

**Writing – review & editing:** Awan Afiaz, Raaj Kishore Biswas.

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
