## [Decision Letter · Decision Letter 0]

14 Jan 2025

PONE-D-24-52692

More Than Just Visits: Timing, Frequency, and Determinants of Effective Antenatal Care in Bangladesh - BDHS 2007 to 2017–18

PLOS ONE

Dear Dr. Jitu,

Thank you for submitting your manuscript to PLOS ONE. After careful consideration, we feel that it has merit but does not fully meet PLOS ONE’s publication criteria as it currently stands. Therefore, we invite you to submit a revised version of the manuscript that addresses the points raised during the review process.

Three review reports have been obtained. Please find these below. 

We look forward to receiving your revised manuscript.

Kind regards,

Muhammad Haroon Stanikzai

Academic Editor

PLOS ONE

Journal Requirements:

For additional information about PLOS ONE ethical requirements for human subjects research, please refer to http://journals.plos.org/plosone/s/submission-guidelines#loc-human-subjects-research .

3. Please note that your Data Availability Statement is currently missing the repository name. If your manuscript is accepted for publication, you will be asked to provide these details on a very short timeline. We therefore suggest that you provide this information now, though we will not hold up the peer review process if you are unable.

5. Please ensure that you refer to Figure 1 and 2 in your text as, if accepted, production will need this reference to link the reader to the figure.

6. We note that Figure 2 in your submission contain map/satellite images which may be copyrighted. All PLOS content is published under the Creative Commons Attribution License (CC BY 4.0), which means that the manuscript, images, and Supporting Information files will be freely available online, and any third party is permitted to access, download, copy, distribute, and use these materials in any way, even commercially, with proper attribution. For these reasons, we cannot publish previously copyrighted maps or satellite images created using proprietary data, such as Google software (Google Maps, Street View, and Earth). For more information, see our copyright guidelines: http://journals.plos.org/plosone/s/licenses-and-copyright.

7. We note you have included a table to which you do not refer in the text of your manuscript. Please ensure that you refer to Table 2 in your text; if accepted, production will need this reference to link the reader to the Table.

Additional Editor Comments:

- Please revise categories for media exposure (yes vs no).

- Please revise categories for mobile ownership (yes vs no).

- It would be desirable for the authors to mention in introduction why it is important for the population of women studied in Bangladesh to analyze access to mobile phones.

- Line 264-266: Suggest to add citation: https://journals.plos.org/plosone/article?id=10.1371/journal.pone.0309300

Reviewers' comments:

Reviewer's Responses to Questions

**Comments to the Author**

1. Is the manuscript technically sound, and do the data support the conclusions?

Reviewer #1: Yes

Reviewer #2: Yes

Reviewer #3: Yes

2. Has the statistical analysis been performed appropriately and rigorously? 

Reviewer #1: No

Reviewer #2: Yes

Reviewer #3: Yes

3. Have the authors made all data underlying the findings in their manuscript fully available?

Reviewer #1: Yes

Reviewer #2: Yes

Reviewer #3: Yes

4. Is the manuscript presented in an intelligible fashion and written in standard English?

Reviewer #1: Yes

Reviewer #2: Yes

Reviewer #3: Yes

5. Review Comments to the Author

Reviewer #1: Peer Review Report

Title: "More Than Just Visits: Timing, Frequency, and Determinants of Effective Antenatal Care in Bangladesh - BDHS 2007 to 2017-18"

General Assessment:

This manuscript presents a comprehensive analysis of antenatal care (ANC) utilization patterns in Bangladesh using nationally representative data from two time points. The study makes a valuable contribution to understanding the factors influencing both the timing and frequency of ANC visits, with important implications for maternal health policy and practice.

Strengths:

The study demonstrates several notable strengths through its approach to addressing an important public health issue with clear policy implications. The use of nationally representative data from two time points (2007 and 2017-18) allows for meaningful temporal comparison and trend analysis. The statistical methodology is robust, employing multiple analytical approaches including spatial mapping and classification trees. The theoretical framework is well-conceived and appropriately applied to the research context.

Major Concerns:

Methodological Issues:

The authors need to better justify their choice of cutoff points for "late" ANC initiation. While they follow national guidelines, a discussion of alternative definitions used in the literature would strengthen the paper. The potential for recall bias in self-reported data should be more thoroughly addressed in the limitations section.

Analytical Concerns:

The classification tree analysis, while innovative, requires more detailed explanation of the methodology and interpretation criteria. The spatial analysis would benefit from statistical tests of spatial autocorrelation to support the observed patterns.

Results Presentation:

Some figures (particularly Figure 2) need clearer labeling and legend explanations. The tables would benefit from consistent formatting and clearer presentation of confidence intervals.

Technical Corrections:

Throughout the manuscript, several acronyms are not defined at first use. There is inconsistent use of decimal places in reported statistics. Some references need updating to more recent sources to reflect current understanding of the field.

Specific Recommendations:

Introduction:

The rationale for comparing 2007 and 2017-18 specifically needs strengthening. Additional context about global ANC recommendations and how Bangladesh's guidelines differ would enhance the reader's understanding of the study's framework.

Methods:

The authors should provide more detail about the handling of missing data and clarify the criteria for inclusion/exclusion of variables in the regression models. The choice of classification tree parameters requires additional explanation to ensure reproducibility.

Results:

Effect sizes should be included alongside p-values consistently throughout the results section. The authors should expand their analysis to include more detailed subgroup analyses for urban/rural differences. Sensitivity analyses for different definitions of "late" ANC would strengthen the robustness of the findings.

Discussion:

The policy implications of the findings need expansion, particularly regarding practical implementation strategies. The comparison with other low- and middle-income countries should be strengthened to provide broader context. The sustainability of observed improvements requires more thorough examination.

Major revision is recommended. While the core analysis is sound and the findings are important, several methodological and presentation issues need to be addressed before publication. The authors should carefully consider each point raised in this review and provide detailed responses to strengthen the manuscript's contribution to the field.

Reviewer #2: Dear Editors and Authors,

Thank you for giving me the opportunity to review the manuscript, titled “More Than Just Visits: Timing, Frequency, and Determinants of Antenatal Care in Bangladesh - BDHS 2007 to 2017–18”.

The authors of the study chose a very important topic related to maternal health in Bangladesh. This study examined the factors contributing to delayed ANC and <4 ANC visits, and the association between delayed ANC and <4 ANC. The authors used data from Bangladesh DHS-2007 and DHS2017–18 from women of reproductive age. Geospatial mapping and logistic regression were used to study special pattern and to identify factors associated with delayed ANC and <4 ANC visits. The authors found that late ANC was associated with a significant increase in the odds of <4 ANC visits both in 2007 and in 2017–18.

My assessment of the manuscript is that it is well written. The findings from this study have the potential to impact health interventions and policy to improve initiation of timely ANC visit and use of ANC services in Bangladesh and developing countries. The authors need to address the following issues before the manuscript can be considered for publication.

Throughout the manuscript, the authors should be specified with <4 ANC vs. ≥4 ANC, and should not use the term “adequate or adequacy or inadequacy” because currently WHO recommends ≥8 ANC visits and I don’t think 4-7 ANC visits can be considered adequate based on the revised WHO’s guide.

Abstract

• The purpose of the sentence “… while also evaluating the impact of late ANC initiation on the overall inadequacy of ANC visits.” is not clear, because this study did not examine the impact of late ANC on inadequacy of ANC; but rather examined their associations. This needs to be revised.

• The authors should define the two outcomes of “late ANC, and “<4 ANC visits” in the abstract

• The authors should clarify that they studied the association between the two outcomes, using logistic regression, and provided the ORs of 4.60 for 2007, and 4.68 for 2017-2018, with 95%CIs.

Theoretical framework

• The authors state in lines 101-103 that their analysis explored “accessibility, acceptability, affordability, and awareness of ANC visits” and the data constraints limited the exploration of the dimension of availability”. I’m not sure if this is correct to say all these dimensions. The authors need to state the aspects they actually examined in this study. For example, they did not examine the affordability dimension.

Results

• In Table 1, the authors need to add “total number” for late ANC, and for <4 ANC. The presentation for descriptive statistics is normally column %, and this should be here as well. Currently the percentage for the respective variable becomes 100% only for total sample in 2007 and in 2017-18. This should be the case for the total number for late ANC and for <4 ANC.

• In Table 2, the authors need to add “total number” for late ANC and for <4 ANC. They should also use the term late ANC and <4 ANC, instead of timing of ANC and number of ANC, because they did not use them as continuous variables, but rather binary variables. Also results of 2007 should be presented first in the column, then results from 2017-2018 (the same issue is there in Table 3).

Reviewer #3: the authors have to brings some minor changes to increase the quality of their manuscripts.

they used appropriate statistical tests and the tables are designed well.

the authors mentioned that all the data are available if its need it.

6. PLOS authors have the option to publish the peer review history of their article (what does this mean? ). If published, this will include your full peer review and any attached files.

**Do you want your identity to be public for this peer review?** For information about this choice, including consent withdrawal, please see our Privacy Policy .

Reviewer #1: No

Reviewer #2: **Yes: ** Essa Tawfiq

Reviewer #3: No

---

## [Author Response · Author response to Decision Letter 1]

27 Feb 2025

28 February 2024

Muhammad Haroon Stanikzai

Academic Editor

PLOS ONE

RE: PLOS ONE Decision: Revision required [PONE-D-24-52692] - [EMID:989dff884a36553c]

Dear Editor,

Thank you for proving us the opportunity to revise the paper titled “More Than Just Visits: Timing, Frequency, and Determinants of Effective Antenatal Care in Bangladesh - BDHS 2007 to 2017–18” for publication in PLOS ONE.

We have addressed the reviewers’ comments in our responses and incorporated related changes into the original manuscript through a revision. For convenience, the reviewers’ comments are highlighted in blue, and our responses are in black font.

The primary changes made in the revised version of the manuscript include:

Revised methodology section including variable construction, data description and classification tree method description.

Revised results section by updating table format included p-value alongside effect size as requested by the reviewer.

Revised discussion section including current literature/references to make comparison of our findings with research conducted in other LMICs and adding policy relevance of our findings.

In addition to the above, all authors have read the revised manuscript. Some minor changes throughout the revised version of the manuscript are made, mostly to improve clarity of expressions and correct grammatical errors.

We would like to thank you and reviewers for all the valuable feedback and suggestions.

Sincere regards,

The authors.

Title: More Than Just Visits: Timing, Frequency, and Determinants of Effective Antenatal Care in Bangladesh - BDHS 2007 to 2017–18

Manuscript ID: PONE-D-24-52692

Journal Requirements:

Please ensure that your manuscript meets PLOS ONE's style requirements, including those for file naming. The PLOS ONE style templates can be found at https://journals.plos.org/plosone/s/file?id=wjVg/PLOSOne_formatting_sample_main_body.pdf and https://journals.plos.org/plosone/s/file?id=ba62/PLOSOne_formatting_sample_title_authors_affiliations.pdf

Authors’ response

Thank you for raising the point. We have revised the file names and ensured that our manuscript meets PLOS ONE's style requirements.

Please provide additional details regarding participant consent. In the ethics statement in the Methods and online submission information, please ensure that you have specified (1) whether consent was informed and (2) what type you obtained (for instance, written or verbal, and if verbal, how it was documented and witnessed). If your study included minors, state whether you obtained consent from parents or guardians. If the need for consent was waived by the ethics committee, please include this information.

For additional information about PLOS ONE ethical requirements for human subjects research, please refer to http://journals.plos.org/plosone/s/submission-guidelines#loc-human-subjects-research.decision??

Authors’ response

Thanks for your interest on this important issue. This work does not include any studies with human participants conducted by the authors. Therefore, no ethical approval was needed for our study as we are using secondary data sets provided by the Demographic Health Survey (DHS). The survey protocol for Bangladesh DHS (BDHS) datasets utilized in this study received approval by institutional review boards (IRBs) at ICF and the Bangladesh Medical Research Council (BMRC). Written informed consent was obtained from each participant before conducting the interviews. Participants who did not provide consent were excluded from the survey. Participants were made aware that their participation was entirely voluntary. They were assured that their responses would be kept confidential and anonymous. Additionally, they were informed of their right to skip any questions they preferred not to answer and could withdraw from the interview at any point. All data were fully anonymized before the authors were given access. We have updated the Ethical consideration statement accordingly (page 7 of the revised manuscript).

“Ethical Considerations

This work does not include any studies with human participants conducted by the authors. Therefore, no ethical approval was needed for our study as we are using secondary data sets provided by the Demographic Health Survey (DHS). The survey protocol for the Bangladesh DHS (BDHS) datasets utilized in this study received approval by institutional review boards (IRBs) at ICF and the Bangladesh Medical Research Council (BMRC). Written informed consent was obtained from each participant before conducting the interviews. Participants who did not provide consent were excluded from the survey. Participants were made aware that their participation was entirely voluntary. They were assured that their responses would be kept confidential and anonymous. Additionally, they were informed of their right to skip any questions they preferred not to answer and could withdraw from the interview at any point. All data were fully anonymized before the authors were given access. The datasets are available on request on the Demographic and Health Survey (DHS) website free of cost for research purposes (https://dhsprogram.com/).”

Please note that your Data Availability Statement is currently missing the repository name. If your manuscript is accepted for publication, you will be asked to provide these details on a very short timeline. We therefore suggest that you provide this information now, though we will not hold up the peer review process if you are unable.

Authors’ response

Thank you. We have updated it form the “Data Availability” field of the submission form (via “Edit Submission”).

“The secondary data sets analyzed during the current study are freely available upon request from the DHS website at https://dhsprogram.com/data/available-datasets.”

Please include your full ethics statement in the ‘Methods’ section of your manuscript file. In your statement, please include the full name of the IRB or ethics committee who approved or waived your study, as well as whether or not you obtained informed written or verbal consent. If consent was waived for your study, please include this information in your statement as well.

Authors’ response

Thank you for raising this point. Ethical approval was not needed for our study since we are using secondary datasets form Bangladesh Demographic and Health Survey (BDHS). “Secondary data vary in terms of the amount of identifying information in it. If the data has no identifying information or is completely devoid of such information or is appropriately coded so that the researcher does not have access to the codes, then it does not require a full review by the ethical board.” (Tripathy, 2013). DHS datasets are popular all over the worlds and there is no identifying information in it, thus it’s not required for us to obtain any additional ethical approval form any IRBs. Besides there are a many existing studies that used this BDHS secondary data sets and published their papers PLOS ONE without further ethical approval for secondary data (Akter et al., 2024; Afiaz et al., 2020; Mansur et al., 2021; Bhowmik et al., 2021).

References:

Akter, S., Hosen, M. S., Khan, M. S., & Pal, B. (2024). Assessing the pattern of key factors on women’s empowerment in Bangladesh: Evidence from Bangladesh Demographic and Health Survey, 2007 to 2017–18. Plos one, 19(3), e0301501.

Afiaz, A., Biswas, R. K., Shamma, R., & Ananna, N. (2020). Intimate partner violence (IPV) with miscarriages, stillbirths and abortions: Identifying vulnerable households for women in Bangladesh. PLoS One, 15(7), e0236670.

Mansur, M., Afiaz, A., & Hossain, M. S. (2021). Sociodemographic risk factors of under-five stunting in Bangladesh: Assessing the role of interactions using a machine learning method. Plos one, 16(8), e0256729.

Bhowmik, J., Biswas, R. K., Williams, J., & Dey, S. R. (2024). Women's decision‐making power can influence modern contraceptive use: Evidence from Bangladesh. The International Journal of Health Planning and Management, 39(5), 1503-1515.

Tripathy, J. P. (2013). Secondary data analysis: Ethical issues and challenges. Iranian journal of public health, 42(12), 1478.

Please ensure that you refer to Figure 1 and 2 in your text as, if accepted, production will need this reference to link the reader to the figure.

Authors’ response

Thank you for the point. We have referred Figure 1 and Figure 2 in the text as follows (page 5 and page 16 of the revised manuscript),

“While our analysis explores aspects such as accessibility, acceptability, affordability, and awareness of ANC visits, data constraints limit the exploration of the dimension of availability (Fig 1).”

“The map presented in Fig 2 illustrates the regional distribution of late and low (< 4) ANC visits in Bangladesh.”

We note that Figure 2 in your submission contain map/satellite images which may be copyrighted. All PLOS content is published under the Creative Commons Attribution License (CC BY 4.0), which means that the manuscript, images, and Supporting Information files will be freely available online, and any third party is permitted to access, download, copy, distribute, and use these materials in any way, even commercially, with proper attribution. For these reasons, we cannot publish previously copyrighted maps or satellite images created using proprietary data, such as Google software (Google Maps, Street View, and Earth). For more information, see our copyright guidelines: http://journals.plos.org/plosone/s/licenses-and-copyright.

Authors’ response

Apologies for any confusion. Figure 2 of our manuscript contain spatial mappings. This map is not copyrighted. This is original image created by us using the statistical software R. The R packages and shapefile used in this manuscript are listed on page 7; in the last paragraph of the Statistical analyses section. Besides, we did not use any R packages that source their maps from Google Maps. Instead, we utilized the administrative boundary shapefile of South Asian countries from an openly licensed database, the geoBoundaries Global Administrative Database (Runfola et al., 2020), which we have cited in the last paragraph of the Statistical analyses section.

Reference:

Runfola D, Anderson A, Baier H, Crittenden M, Dowker E, Fuhrig S, Goodman S, Grimsley G, Layko R, Melville G, Mulder M. geoBoundaries: A global database of political administrative boundaries. PloS one. 2020 Apr 24;15(4):e0231866.

We note you have included a table to which you do not refer in the text of your manuscript. Please ensure that you refer to Table 2 in your text; if accepted, production will need this reference to link the reader to the Table.

Authors’ response

Thank you for noticing this. We have updated our manuscript and referred Table 2 and, in the text, as follows (page 11 of revised manuscript),

“The results of binary logistic regression model adjusted for sociodemographic factors with timing of first ANC visit and number of ANC visits (low (< 4) ANC visits) as outcome is shown in Table 2.”

Please include captions for your Supporting Information files at the end of your manuscript, and update any in-text citations to match accordingly. Please see our Supporting Information guidelines for more information: http://journals.plos.org/plosone/s/supporting-information.

Authors’ response

Thank you for raising the point. We have updated the manuscript accordingly and the following section has been added at the end of our manuscript (page 25 of the revised manuscript),

“Supporting information

S1 Table. The frequency distribution of late ANC and low (< 8) ANC visits by sociodemographic factors.

S2 Table. The frequency distribution of low (<8) ANC visits by timing of first ANC visits.

S3 Table. Binary logistic regression model adjusted for sociodemographic factors and timing of first ANC visit with number of ANC visits (low (<4) ANC visits) as outcome.

S4 Table. Binary logistic regression model adjusted for sociodemographic factors with timing of first ANC visit and number of ANC visits (low (<8) ANC visits) as outcome.

S5 Table. Binary logistic regression model adjusted for sociodemographic factors and timing of first ANC visit with number of ANC visits (low (<8) ANC visits) as outcome.

S6 Table. GVIF for binary logistic regression model adjusted for sociodemographic factors with timing of first ANC visit as outcome.

S7 Table. GVIF for binary logistic regression model adjusted for sociodemographic factors with number of ANC visits (low (<4) ANC visits) as outcome.

S8 Table. GVIF for binary logistic regression model adjusted for sociodemographic factors and timing of first ANC visit with number of ANC visits (low (<4) ANC visits) as outcome

S9 Table. GVIF for binary logistic regression model adjusted for sociodemographic factors number of ANC visits (low (<8) ANC visits) as outcome.

S10 Table. GVIF for binary logistic regression model adjusted for

---

## [Decision Letter · Decision Letter 1]

10 Mar 2025

More Than Just Visits: Timing, Frequency, and Determinants of Effective Antenatal Care in Bangladesh - BDHS 2007 to 2017–18

PONE-D-24-52692R1

Dear Dr. Jitu,

We’re pleased to inform you that your manuscript has been judged scientifically suitable for publication and will be formally accepted for publication once it meets all outstanding technical requirements.

Kind regards,

Muhammad Haroon Stanikzai

Academic Editor

PLOS ONE

Additional Editor Comments (optional):

Thank you for addressing reviewers' comments.

Reviewers' comments:

Reviewer's Responses to Questions

**Comments to the Author**

1. If the authors have adequately addressed your comments raised in a previous round of review and you feel that this manuscript is now acceptable for publication, you may indicate that here to bypass the “Comments to the Author” section, enter your conflict of interest statement in the “Confidential to Editor” section, and submit your "Accept" recommendation.

Reviewer #1: All comments have been addressed

Reviewer #2: All comments have been addressed

Reviewer #3: All comments have been addressed

2. Is the manuscript technically sound, and do the data support the conclusions?

Reviewer #1: Yes

Reviewer #2: Yes

Reviewer #3: Yes

3. Has the statistical analysis been performed appropriately and rigorously? 

Reviewer #1: Yes

Reviewer #2: Yes

Reviewer #3: Yes

4. Have the authors made all data underlying the findings in their manuscript fully available?

Reviewer #1: Yes

Reviewer #2: Yes

Reviewer #3: Yes

5. Is the manuscript presented in an intelligible fashion and written in standard English?

Reviewer #1: Yes

Reviewer #2: Yes

Reviewer #3: Yes

6. Review Comments to the Author

Reviewer #1: Many thanks to the authors as all my comments have been addressed. The manuscript is technically sound now and I have no further comments.

Reviewer #2: Dear Authors and Editors,

Thank you for sharing this article with me for a peer review. All comments I raised in my previous review of the article have been adequately addressed in this revised version.

Kind regards,

Essa Tawfiq

Reviewer #3: (No Response)

7. PLOS authors have the option to publish the peer review history of their article (what does this mean? ). If published, this will include your full peer review and any attached files.

**Do you want your identity to be public for this peer review?** For information about this choice, including consent withdrawal, please see our Privacy Policy .

Reviewer #1: No

Reviewer #2: No

Reviewer #3: No

---

## [Editor Report · Acceptance letter]

PONE-D-24-52692R1

PLOS ONE

Dear Dr. Jitu,

I'm pleased to inform you that your manuscript has been deemed suitable for publication in PLOS ONE. Congratulations! Your manuscript is now being handed over to our production team.

Kind regards,

on behalf of

Dr. Muhammad Haroon Stanikzai

Academic Editor

PLOS ONE